# Glucocorticoid signaling in pancreatic islets modulates gene regulatory programs and genetic risk of type 2 diabetes

Anthony Aylward[1][◔], Mei-Lin Okino[2][◔], Paola Benaglio[2], Joshua Chiou[3], Elisha Beebe[2], Jose Andres Padilla[2], Sharlene Diep[2], Kyle J. Gaulton[2,4]*

1 Bioinformatics and Systems Biology graduate program, University of California San Diego, La Jolla, California, United States of America, 2 Department of Pediatrics, University of California San Diego, La Jolla, California, United States of America, 3 Biomedical Sciences graduate program, University of California San Diego, La Jolla, California, United States of America, 4 Institute for Genomic Medicine, University of California San Diego, La Jolla, California, United States of America

◔ These authors contributed equally to this work.
* kgaulton@ucsd.edu

**Data Availability Statement:** The authors confirm that all data underlying the findings are fully available without restriction. All raw data are available from the GEO database (GSE167250). All

## Abstract

Glucocorticoids are key regulators of glucose homeostasis and pancreatic islet function, but the gene regulatory programs driving responses to glucocorticoid signaling in islets and the contribution of these programs to diabetes risk are unknown. In this study we used ATAC-seq and RNA-seq to map chromatin accessibility and gene expression from eleven primary human islet samples cultured *in vitro* with the glucocorticoid dexamethasone at multiple doses and durations. We identified thousands of accessible chromatin sites and genes with significant changes in activity in response to glucocorticoids. Chromatin sites up-regulated in glucocorticoid signaling were prominently enriched for glucocorticoid receptor binding sites and up-regulated genes were enriched for ion transport and lipid metabolism, whereas down-regulated chromatin sites and genes were enriched for inflammatory, stress response and proliferative processes. Genetic variants associated with glucose levels and T2D risk were enriched in glucocorticoid-responsive chromatin sites, including fine-mapped variants at 51 known signals. Among fine-mapped variants in glucocorticoid-responsive chromatin, a likely casual variant at the 2p21 locus had glucocorticoid-dependent allelic effects on beta cell enhancer activity and affected *SIX2* and *SIX3* expression. Our results provide a comprehensive map of islet regulatory programs in response to glucocorticoids through which we uncover a role for islet glucocorticoid signaling in mediating genetic risk of T2D.

## Author summary

Glucocorticoids regulate inflammation and metabolism and are widely used in the treatment of immune disorders, although prolonged exposure to glucocorticoids can lead to the development of diabetes. In this study we determined the response of primary pancreatic islets, which are central to the development of diabetes, to the glucocorticoid dexamethasone at multiple doses and durations. We observed widespread changes in

data underlying graphs are provided in the main text or as Supporting Information.

**Funding:** This work was supported by National Institute of Diabetes and Digestive and Kidney Diseases awards DK114650, DK122607, and DK120429 to KJG. This publication includes data generated at the UC San Diego IGM Genomics Center utilizing an Illumina NovaSeq 6000 that was purchased with funding from a National Institutes of Health SIG grant (#S10 OD026929). The funders had no role in study design, data collection and analysis, decision to publish, or preparation of the manuscript.

pancreatic islets after glucocorticoid treatment at glucocorticoid receptor binding sites, as well as at key genes involved in islet function and processes related to steroid and lipid metabolism, ion channel activity, inflammation, and growth. Genetic variants affecting type 2 diabetes and glucose levels were located in sites affected by glucocorticoids at many genomic regions, and highlighted genes regulated by these sites through which glucocorticoid signaling may contribute directly to the development of diabetes. Together these results provide key insight into how glucocorticoid treatment affects pancreatic islet function and risk of diabetes.

## Introduction

Glucocorticoids are steroid hormones produced by the adrenal cortex which broadly regulate inflammatory, metabolic and stress responses and are widely used in the treatment of immune disorders [1–3]. The metabolic consequences of glucocorticoid action are directly relevant to diabetes pathogenesis, as chronic glucocorticoid exposure causes hyperglycemia and steroid-induced diabetes and endogenous excess of glucocorticoids causes Cushing's syndrome in which diabetes is a common co-morbidity [4,5]. Glucocorticoids contribute to the development of diabetes both through insulin resistance and obesity via effects on adipose, liver and muscle, as well as through pancreatic islet dysfunction [4]. In islets, glucocorticoid signaling has been shown to modulate numerous processes such as insulin secretion, ion channel activity, cAMP signaling, proliferation and development [6–11].

The effects of glucocorticoids on cellular function are largely mediated through regulation of transcriptional activity. Glucocorticoids diffuse through the cell membrane into cytoplasm and bind the glucocorticoid receptor (GR), which is then translocated into the nucleus where it binds DNA and modulates the transcriptional program [12–15]. Gene activity can be affected by GR via direct genomic binding and regulation as well as indirectly through physical interaction with other transcriptional regulators [13–17]. Previous studies have profiled glucocorticoid signaling by mapping genomic locations of GR binding and other epigenomic features such as histone modifications and chromatin accessibility in response to endogenous glucocorticoids such as cortisol or analogs such as dexamethasone [13,14,18,19]. Studies have also shown that the genomic function of GR is largely mediated via binding to regions of accessible chromatin [20,21].

Genetic studies have identified hundreds of genomic loci that contribute to diabetes risk and which primarily map to non-coding sequence and affect gene regulation [22–25]. Risk variants for type 2 diabetes (T2D) are enriched for pancreatic islet regulatory sites [22–24,26,27], while type 1 diabetes (T1D) risk variants are enriched for immune cell as well as islet regulatory sites. The specific mechanisms of most risk variants in islets are unknown, however, which is critical for understanding the genes and pathways involved in disease pathogenesis and for the development of novel therapeutic strategies. Previous studies of islet chromatin have focused predominantly on normal, non-disease states [27–33], although recent evidence has shown that diabetes risk variants can interact with environmental stimuli to affect islet chromatin and gene regulatory programs [34].

The effects of glucocorticoid and other steroid hormone signaling on islet regulatory programs and how these signals interact with diabetes risk variants, however, are largely unknown. In this study we profiled islet accessible chromatin and gene expression in primary human pancreatic islets exposed *in vitro* to the glucocorticoid dexamethasone. Glucocorticoid signaling had widespread effects on islet accessible chromatin and gene expression levels. Up-

regulated chromatin sites were strongly enriched for glucocorticoid receptor binding and up-regulated genes were enriched for processes related to ion channel activity and steroid and lipid metabolism. Conversely, down-regulated sites and genes were involved in inflammation, stress response and proliferation. Genetic variants affecting T2D risk and glucose levels were significantly enriched in glucocorticoid-responsive chromatin sites, including a likely causal variant at the *SIX2/3* locus which had glucocorticoid-dependent effects on beta cell enhancer activity and affected *SIX2* and *SIX3* expression. Together our results provide a comprehensive map of islet gene regulatory programs in response to glucocorticoids which will facilitate a greater mechanistic understanding of glucocorticoid signaling and its role in islet function and diabetes risk.

## Results

### Map of gene regulation in pancreatic islets in response to glucocorticoid signaling

In order to determine the effects of glucocorticoid signaling on pancreatic islet regulation, we cultured primary islet cells *in vitro* with dexamethasone at several different doses (100 ng/mL for 24hr, 4 ng/mL for 6hr and 24hr) as well as in untreated conditions and measured accessible chromatin and gene expression levels in both treated and untreated cells. An overview of the study design is provided in Fig 1A.

We assayed gene expression in dexamethasone-treated and untreated islets from 6 total samples using RNA-seq (S1 Table; see Methods). Across replicate samples we observed changes in expression levels of genes both known to be induced by dexamethasone such as *ZBTB16* [35–37] and *VIPR1* [38] as well as those suppressed by dexamethasone such as *IL11* [39] in both the high-dose (100 ng/mL) and low-dose (4 ng/mL) treatments (Figs 1B, 1C, S1A, S1B and S1C). We next assayed accessible chromatin in dexamethasone-treated and untreated islets from 9 total samples using ATAC-seq (S1 Table; see Methods). Across replicate samples we observed reproducible changes in islet accessible chromatin signal concordant with changes in gene expression. For example, accessible chromatin signal was notably induced at several sites proximal to the *ZBTB16* and *VIPR1* genes in dexamethasone-treated compared to untreated islets in both high- and low-dose treatments (Figs 1D, 1E, S2, S3 and S4). Similarly, accessible chromatin signal was reduced at a site proximal to the *IL11* promoter in glucocorticoid-treated compared to untreated islets (S5 Fig).

### Islet accessible chromatin sites with differential activity in response to glucocorticoid signaling

To understand the effects of glucocorticoid signaling on accessible chromatin in islets at a genome-wide level, we first performed principal components analysis (PCA) using normalized read counts in chromatin sites for each treated and untreated islet ATAC-seq sample (see Methods). We observed reproducible differences in accessible chromatin profiles in dexamethasone-treated compared to untreated islets across replicate samples, where the effects of low-dose treatment (4 ng/mL, n = 3) were intermediate to high-dose treatment (100 ng/mL, n = 6) relative to untreated samples (n = 9) (Fig 2A).

We then identified specific islet accessible chromatin sites with significant differential activity in glucocorticoid treatment compared to untreated control cells. We first defined a canonical set of 127,228 islet accessible chromatin sites genome-wide by comparing replicate samples using IDR (see Methods, S2 Table). Among these canonical sites, there were 2,688 sites with significant evidence (FDR < .10) for differential activity in glucocorticoid signaling at high-

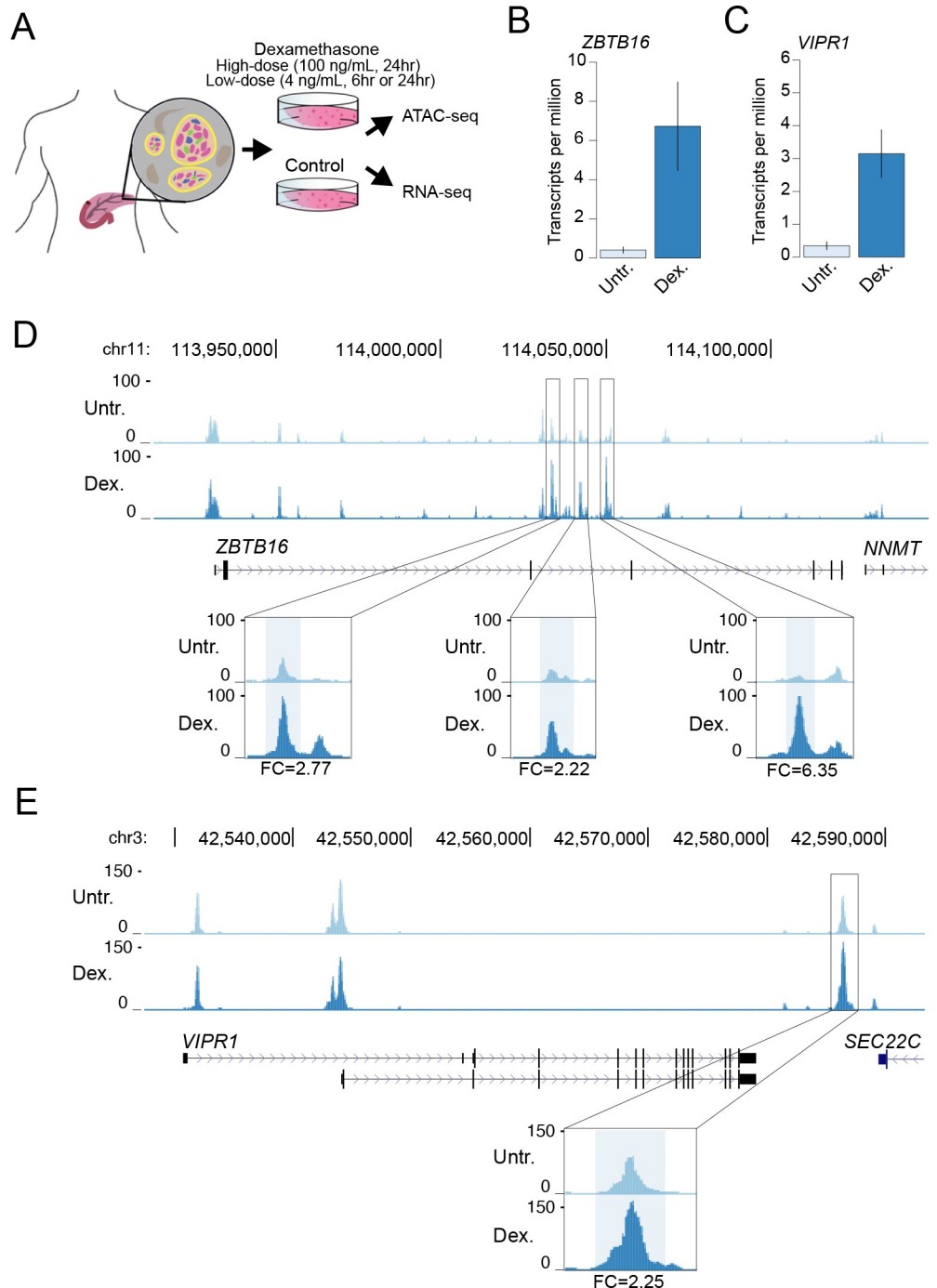

**Fig 1. A map of gene regulation in pancreatic islets in response to glucocorticoid signaling.** (A) Overview of study design. Primary pancreatic islet samples were split and separately cultured in normal conditions and including the glucocorticoid dexamethasone at either a high-dose (100ng/mL for 24hr) or low-dose (4 ng/mL for 6hr or 24hr) treatment, and then profiled for gene expression and accessible chromatin using RNA-seq and ATAC-seq assays. Genes with known induction in glucocorticoid signaling (B) *ZBTB16* and (C) *VIPR1* had increased expression in glucocorticoid-treated islets compared to untreated islets. Values represent mean and standard error. (D) At the *ZBTB16* locus several accessible chromatin sites intronic to *ZBTB16* had increased accessibility in glucocorticoid treated (Dex.) compared to untreated (Untr.) islets. (E) At the *VIPR1* locus an accessible chromatin site downstream of *VIPR1* had increased accessibility in glucocorticoid treated (Dex.) compared to untreated (Untr.) islets. Values in D and E represent RPKM normalized ATAC-seq read counts. Fold-change (FC) in accessible chromatin signal in glucocorticoid treatment compared to untreated indicated at highlighted sites. All results shown are for the high-dose treatment.

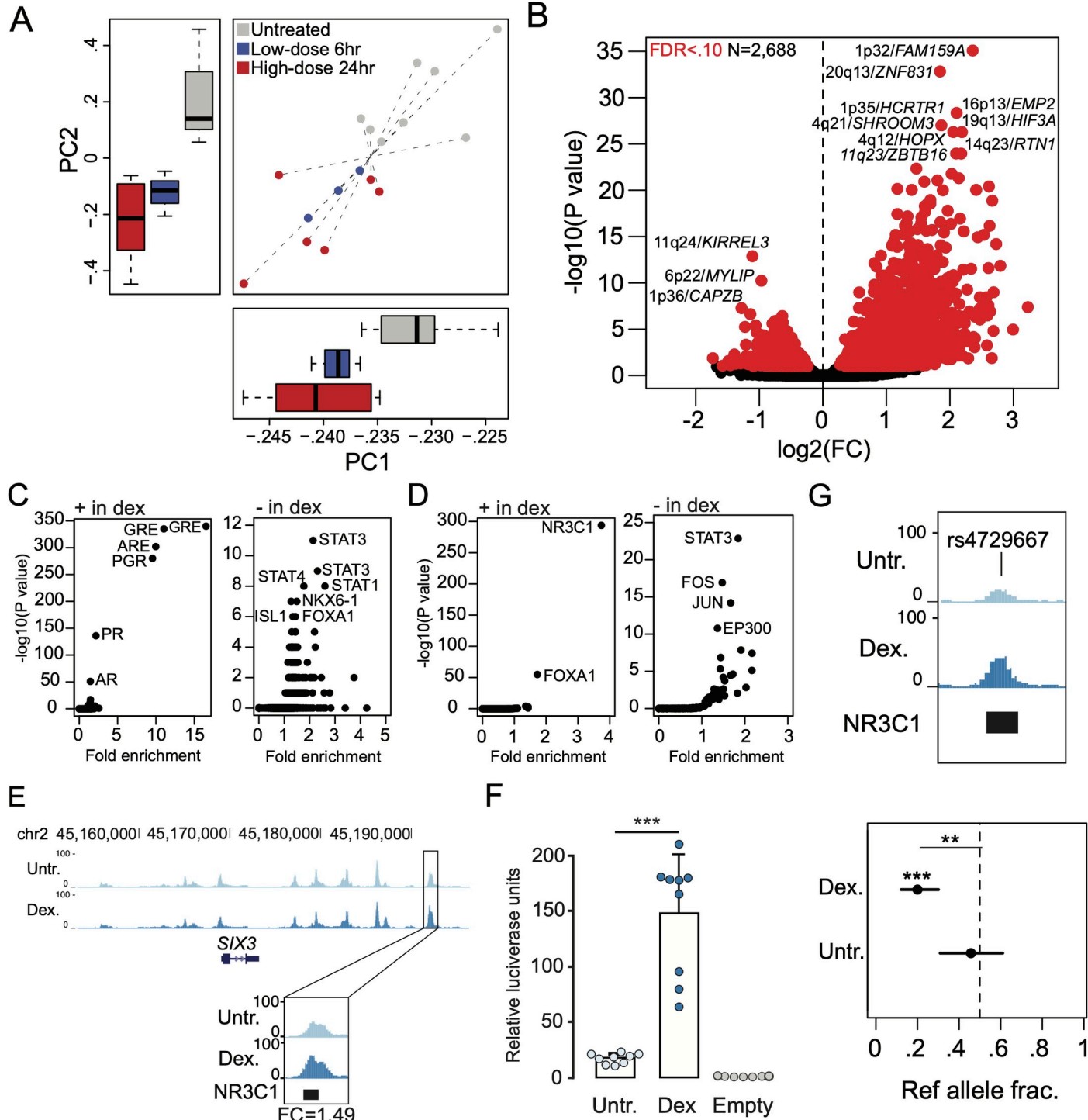

**Fig 2. Glucocorticoid signaling affects chromatin accessibility in pancreatic islets.** (A) Principal components plot showing ATAC-seq signal for high-dose (red) and low-dose (blue) glucocorticoid-treated islets and untreated (grey) islets from 9 total donors. Dashed lines connect assays from the same sample, and box plots on each axis represent the distribution of principal components of samples for each condition. (B) Volcano plot of sites with differential chromatin accessibility in glucocorticoid treated compared to untreated islets. Sites with significant differential activity (FDR < .10) are highlighted in red. The sites with the most significant changes are labelled with the locus and the nearest gene. (C) Transcription factor (TF) sequence motifs enriched in differential chromatin sites with increased activity (+ in dex) and decreased activity (- in dex) in glucocorticoid-treated islets. (D) Enrichment of ChIP-seq sites from ENCODE for 160 TFs in differential chromatin sites with increased activity (+ in dex) and decreased activity (- in dex) in glucocorticoid-treated islets. (E) A chromatin site at the *SIX2/3* locus had increased activity in glucocorticoid-treated islets and overlapped a ChIP-seq site for the glucocorticoid receptor (GR/NR3C1) (top). Fold-change (FC) in accessible chromatin signal in glucocorticoid treatment compared to untreated indicated at the highlighted site for high-dose treatment. (F) The differential site at *SIX2/3* had glucocorticoid-

dependent effects on enhancer activity in gene reporter assays in MIN6 cells (bottom). Values represent mean and standard deviation. (G) Variant rs4729667 mapped in a chromatin site with increased activity in glucocorticoid-treated islets and had stronger allelic imbalance in chromatin accessibility in glucocorticoid-treated compared to untreated islets. Values represent ref allele fraction and 95% confidence intervals. For panels B, C and D the values shown are from results using high-dose treatment. **$P < .01$, ***$P < 1 \times 10^{-4}$.

dose treatment (Fig 2B and S3 Table). Among these 2,688 glucocorticoid-responsive sites, 1,992 had up-regulated activity and 695 had down-regulated activity in glucocorticoid treated compared to untreated cells (Fig 2B and S3 Table). The majority of sites (95%) with differential activity were already accessible in untreated islets, suggesting that sites induced by glucocorticoid signaling are typically not activated *de novo*. Furthermore, a majority of differentially accessible sites (2,453, 91%) were not proximal to promoter regions, suggesting they act via distal regulation of gene activity. At low-dose treatment, 373 sites had differential activity (FDR < .10) in glucocorticoid signaling, where the majority (350) were up-regulated (S3 Table). Among sites with differential activity in either treatment, the effects in high- and low-dose were highly concordant (Spearman ρ = .72, P<2.2x10$^{-16}$) (S6A and S6B Fig).

We next characterized transcriptional regulators underlying changes in glucocorticoid-responsive islet chromatin. First, we identified TF motifs enriched in genomic sequence underneath sites up-regulated and down-regulated in glucocorticoid-treated islets (see Methods). The most enriched sequence motifs in up-regulated sites for both high- and low-dose treatment were for glucocorticoid and other steroid hormone response elements (high-dose: GRE P = 1x10$^{-340}$, ARE P = 1x10$^{-302}$, PGR P = 1x10$^{-280}$; low-dose: GRE P = 1x10$^{-73}$, ARE P = 1x10$^{-66}$, PGR P = 1x10$^{-62}$), in addition to lesser enrichment for TFs relevant to islet function (FOXA1: high-dose P = 1x10$^{-5}$, low-dose P = 1x10$^{-3}$) (Fig 2C and S4 Table). Conversely, down-regulated sites in high-dose treatment were most enriched for sequence motifs for STAT TFs (STAT3 P = 1x10$^{-9}$, STAT1 P = 1x10$^{-8}$) followed by TFs involved in islet function (NKX6.1 P = 1x10$^{-7}$, FOXA1 P = 1x10$^{-6}$) (Fig 2C and S4 Table). Next, we determined enrichment of glucocorticoid-responsive chromatin sites for ChIP-seq TF-binding sites previously identified by the ENCODE project. We observed strongest enrichment of up-regulated accessible chromatin sites in both high- and low-dose treatment for glucocorticoid receptor (NR3C1) binding sites (high-dose ratio = 3.7, P = 1.7x10$^{-294}$, low-dose ratio = 5.4, P = 2.1x10$^{-129}$), and less pronounced enrichment for binding sites of FOXA1 (high-dose ratio = 1.7, P = 1.6x10$^{-55}$; low-dose ratio = 2.3, P = 2.3x10$^{-30}$) and other TFs (Fig 2D and S4 Table). Down-regulated sites were most enriched for STAT binding (STAT3 ratio = 2.1, P = 7.6x10$^{-41}$) as well as enhancer binding TFs such as FOS/JUN (FOS ratio = 1.5, P = 2.3x10$^{-17}$; JUN ratio = 1.7, P = 1.3x10$^{-14}$) and P300 (ratio = 1.4, P = 2.9x10$^{-11}$) (Fig 2D and S4 Table).

Accessible chromatin sites with significant up-regulation in glucocorticoid signaling compared to untreated islets included a site that mapped to the *SIX2/SIX3* locus (Fig 2E and S3 Table), which also harbors genetic variants associated with fasting glucose level and risk of T2D. The glucocorticoid-responsive site at this locus also directly overlapped a NR3C1 ChIP-seq site identified by the ENCODE project (Fig 2E). We tested the glucocorticoid-induced site at this locus (high-dose fold-change = 1.49; P = 1.0x10$^{-5}$; low-dose fold-change = 1.51; P = 4.4x10$^{-4}$) for enhancer activity in luciferase gene reporter assays in dexamethasone-treated and untreated MIN6 mouse insulinoma cells. We observed a significant increase in enhancer activity in dexamethasone-treated cells relative to untreated cells (T-test P = 1.65x10$^{-6}$) (Fig 2F), confirming that this site is highly induced in response to glucocorticoid signaling.

Environmental stimuli can interact with genetic variation to affect chromatin accessibility and gene regulation. We therefore determined the effects of genetic variants on islet accessible chromatin in both glucocorticoid-treated and untreated conditions using allelic imbalance

mapping. We performed microarray genotyping of seven islet samples and imputed genotypes into 39M variants (see Methods). For variants overlapping islet chromatin sites we obtained read counts in samples heterozygote for that variant, corrected for mapping bias using WASP and modeled the resulting counts for imbalance using a beta-binomial test. We then identified variants with evidence (FDR < .10) for allelic imbalance in accessible chromatin from either glucocorticoid-treated or untreated islets (S5 Table). Among imbalanced variants, we further identified those with significant differences in allelic effects (FDR < .10) between glucocorticoid-treated and untreated islets (S5 Table, see Methods). For example, variant rs4729667 at 7q22 mapped in a glucocorticoid-responsive site bound by GR and had significantly stronger imbalance in glucocorticoid-treated islets (GC ref frac. = .20, untr. ref frac. = .46; $P = 3.9 \times 10^{-3}$) (Fig 2G and S5 Table). Conversely, variant rs2291583 at 10p12 in a glucocorticoid-responsive site had significantly stronger imbalance in untreated islets (GC ref frac = .39, untr. ref frac. = .28; $P = 9.6 \times 10^{-4}$) (S5 Table).

These results demonstrate that glucocorticoid signaling broadly affects accessible chromatin in islets including sites both up-regulated through glucocorticoid receptor activity and down-regulated through the activity of STAT and other TFs.

## Genes and pathways with differential regulation in islets in response to glucocorticoid signaling

We next sought to determine the effects of glucocorticoid treatment on gene expression levels. We first performed PCA using gene transcript counts from untreated and dexamethasone-treated islet samples at each treatment dose and duration obtained from RNA-seq assays (see Methods). There were again reproducible differences in expression levels across replicate samples, where the effects of low-dose treatment (4 ng/mL at 24hr, n = 3; 4 ng/mL at 6hr, n = 3) were intermediate to high-dose treatment (100 ng/mL at 24hr, n = 6) relative to untreated samples (n = 6) (Fig 3A).

We identified specific genes with differential expression in response to glucocorticoids compared to untreated islet samples using DESeq2 (see Methods). There were 2,837 genes with significant evidence for differential expression (FDR<0.10) in glucocorticoid signaling at high-dose treatment (S6 Table). Among these genes, 1,348 (47%) were up-regulated and 1,489 (53%) were down-regulated in response to glucocorticoids compared to untreated islets (Fig 3B). Genes with the most significant up-regulation included *EDN3* (log2(FC) = 1.44, FDR = $2.42 \times 10^{-81}$), *FAM115C* (log2(FC) = 1.52, FDR = $3.61 \times 10^{-75}$), *METTL7A* (log2(FC) = 1.81, FDR = $4.36 \times 10^{-71}$), *PRR15L* (log2(FC) = 2.20, FDR = $9.55 \times 10^{-62}$), and *CCND3* (log2(FC) = 0.95, FDR = $9.05 \times 10^{-60}$). Conversely, genes with most significant down-regulation included *PCSK1* (log2(FC) = -1.21, FDR = $2.05 \times 10^{-61}$), *KLHL41* (log2(FC) = -1.31, FDR = $8.84 \times 10^{-59}$), *DHRS2* (log2(FC) = -1.41, FDR = $2.19 \times 10^{-49}$) and *CD36* (log2(FC) = -1.21, FDR = $2.41 \times 10^{-49}$) (Fig 3B). At low-dose treatment 775 and 848 genes had differential expression (FDR < .10) at 6hr and 24hr, respectively (S6 Table and S7A and S7B Fig). Among genes differentially expressed in either treatment, the effects in high- and low-dose were highly concordant (24hr low-dose ρ = .91, $P<2.2 \times 10^{-16}$; 6hr low-dose ρ = .86, $P<2.2 \times 10^{-16}$) (S7C and S7D and S7E Fig).

We determined whether changes in gene expression in glucocorticoid signaling were driven through accessible chromatin, by testing for enrichment of glucocorticoid-responsive chromatin sites for proximity to differentially expressed genes. Glucocorticoid-responsive chromatin sites were significantly more likely to map within 100kb of a gene with glucocorticoid-responsive expression compared to other chromatin sites in islets (high-dose: OR = 1.48, $P = 9.9 \times 10^{-20}$; low-dose: OR = 4.91, $P = 6.5 \times 10^{-36}$). We next performed these analyses separately for sites up- and down-regulated in glucocorticoid signaling. There was significant enrichment of sites

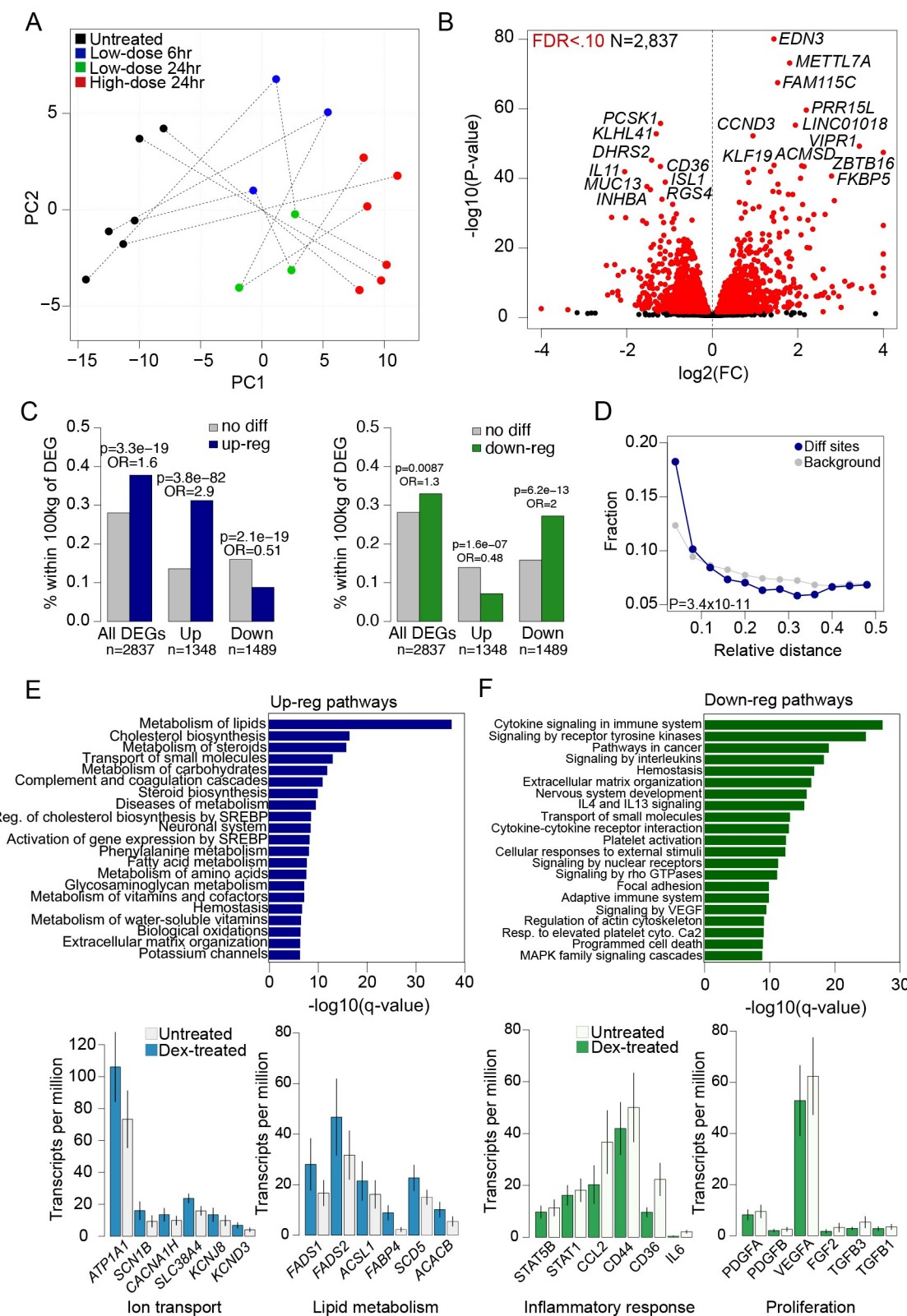

**Fig 3. Glucocorticoid signaling affects gene expression levels in pancreatic islets.** (A) Principal components plot of gene expression from high-dose (red) and low-dose (green 24hr, blue 6hr) glucocorticoid-treated and untreated (black) islets from a total of 6 samples. Dashed lines connect assays from the same sample. (B) Volcano plot showing genes with differential expression in

glucocorticoid-treated islets compared to untreated islets. Genes with significantly differential expression (FDR < .10) are highlighted in red, and genes with pronounced changed in expression are listed. (C) Percentage of accessible chromatin sites with up-regulated activity (left) and down-regulated activity (right) in glucocorticoid-treated islets within 100kb of differentially expressed genes (DEGs) compared to chromatin sites without differential activity. (D) Relative distance metric (from bedtools reldist) between accessible chromatin sites with differential activity (dex) and genes with differential expression compared to all chromatin sites (background). (E) Biological pathway terms enriched among genes with up-regulated expression in glucocorticoid-treated islets (top), and the expression level of selected genes annotated with ion transport and lipid metabolism terms in glucocorticoid-treated and untreated islets (bottom). Values represent mean expression and standard error. (F) Biological pathway terms enriched among genes with up-regulated expression in glucocorticoid-treated islets (top), and the expression level of selected genes annotated with inflammatory response and proliferation pathway terms in glucocorticoid-treated and untreated islets (bottom). Values represent mean expression and standard error. For panels B, C, D and E the values shown are from results using high-dose treatment.

with increased activity in glucocorticoid signaling within 100kb of genes with up-regulated expression specifically (up-reg OR = 2.9, P = $3.8\times10^{-82}$, down-reg OR = 0.51, P = $2.1\times10^{-19}$) (Fig 3C). Similarly, sites with decreased activity in glucocorticoid signaling were enriched within 100kb of genes with down-regulated expression (down-reg OR = 2.0, P = $6.2\times10^{-13}$, up-reg OR = 0.48, P = $1.6\times10^{-7}$) (Fig 3C). Furthermore, we also observed an enrichment of glucocorticoid-responsive chromatin sites for closer proximity to genes with glucocorticoid-responsive expression compared to background sites (Kolmogorov-Smirnov P = $3.4\times10^{-11}$) (Fig 3D).

In order to understand the molecular pathways affected by glucocorticoid activity in islets, we tested genes up- and down-regulated in glucocorticoid signaling for gene set enrichment using pathway and gene ontology (GO) terms (see Methods). Up-regulated genes in high-dose treatment were enriched for gene sets related to steroid metabolism (steroid metabolic process FDR = $8.94\times10^{-30}$), lipid metabolism (lipid biosynthetic process FDR = $1.93\times10^{-32}$), potassium and other ion transport (potassium channels FDR = $5.71\times10^{-7}$; regulation of ion transport FDR = $1.93\times10^{-17}$), and extracellular matrix organization (FDR = $3.68\times10^{-7}$) (Fig 3E and S7 Table). Similar gene sets were enriched among genes up-regulated in low-dose treatments (S7 Table). Numerous genes that function in ion transport were up-regulated in glucocorticoid signaling; for example *ATP1A1*, *SCN1B*, *SCNN1A*, *CACNA1H*, *CACNG4*, *SLC38A4*, *TRPV6* as well as potassium channel genes including *KCNJ2*, *KCNAB1*, *KCNF1*, *KCNJ8*, and *KCND3* (Fig 3E and S6 Table). Up-regulated genes also included numerous that function in lipid metabolism including *FADS1*, *FADS2*, *ACSL1*, *SCD5*, *FABP4*, *ACACB*, and *ANGPTL4* (Fig 3E and S6 Table).

Conversely, genes down-regulated in glucocorticoid signaling were enriched for inflammatory response (cytokine signaling in immune system FDR = $2.2\times10^{-27}$, signaling by interleukins FDR = $9.50\times10^{-19}$), extracellular matrix, cell adhesion and morphogenesis (extracellular matrix organization FDR = $1.53\times10^{-17}$, regulation of cell adhesion FDR = $2.48\times10^{-42}$, cellular component morphogenesis FDR = $2.45\times10^{-37}$), and cell differentiation and proliferation terms (neg. regulation of cell differentiation FDR = $2.18\times10^{-36}$) (Fig 3F and S7 Table). Similar gene sets were enriched among genes down-regulated in low-dose treatments (S7 Table). Down-regulated genes included those involved in the inflammatory response such as *IL6*, *STAT5B*, *STAT3*, *STAT4*, *SMAD3*, *CXCL12*, *CCL2*, *CD44*, *CD36*, *RELB*, *IRF1*, extracellular matrix formation such matrix metalloproteinase genes such as *MMP3*, *MMP7*, *MMP9* and matrix components such as *LAMA4* and *LAMC2*, islet function and pancreatic differentiation such as *ISL1*, *PAX6*, *NKX6-1*, *HES1* and *JAG1*, and proliferation and growth factors such as *PDGFA*, *PDGFB*, *FGF2*, *TGFB3* and *VEGFA* (Fig 3F and S6 Table).

These results demonstrate that glucocorticoid signaling in islets up-regulates genes involved in steroid and lipid metabolism and ion channel activity, and down-regulates key genes in islet function as well as genes involved in inflammation, proliferation and extracellular matrix formation.

## T2D and glucose associated variants map in glucocorticoid-responsive islet chromatin

Genetic variants associated with diabetes risk are enriched in pancreatic islet regulatory elements. As these studies have been performed primarily using non-diabetic donors in normal (untreated) conditions, however, the role of environmental stimuli in modulating diabetes-relevant genetic effects on islet chromatin is largely unknown. We therefore tested for enrichment of diabetes and fasting glycemia associated variants in glucocorticoid-responsive islet chromatin sites using fgwas [40] (see Methods). We observed enrichment of variants influencing T2D risk and blood sugar (glucose) levels in chromatin sites with differential activity in both high- and low-dose glucocorticoid treatment (T2D high-dose ln(enrich) = 3.71, 95% CI = 3.03,4.25; T2D low-dose ln(enrich) = 4.23, 95% CI = 2.66,5.20; blood sugar high-dose ln(enrich) = 3.92, 95% CI = 0.86,5.70; blood sugar low-dose ln(enrich) = 6.20, 95% CI = 3.92,8.42) (Fig 4A). Conversely, we observed no evidence for enrichment of T1D risk variants (high-dose ln(enrich) = -28.00, 95% CI = -48.00,3.39; low-dose ln(enrich) = -23.82, 95% CI = -43.8,5.29) (Fig 4A).

We next catalogued fine-mapped variants overlapping glucocorticoid-responsive islet chromatin using 99% credible sets of T2D and glucose level signals from DIAMANTE and Biobank Japan (BBJ) [22,41] (see Methods). We identified 126 fine-mapped variants at 51 signals that overlapped a glucocorticoid-responsive site (S8 Table). We further identified 511 variants genome-wide in glucocorticoid-responsive sites with at least nominal evidence for T2D association (P < .005) in DIAMANTE or BBJ GWAS (S8 Table). We prioritized potential target genes of T2D- and glucose-associated variants in glucocorticoid-responsive chromatin by identifying genes proximal to these sites with differential expression. For example, T2D-associated variants at the 11q12 locus mapped in a site induced by glucocorticoids proximal to *SCD5* and *TMEM150C* which both had up-regulated expression (Fig 4B and S3 and S8 Tables). Similarly, T2D-associated variants at the 4q31 locus mapped in a site down-regulated in glucocorticoids proximal to *FBXW7* which had down-regulated expression (S7A Fig and S3 and S8 Tables). Outside of known T2D loci we observed additional examples such as at the 7p15 locus where rs1107376 (T2D P = 2.2x10⁻⁴) mapped in a glucocorticoid-induced site proximal to *NPY* which had glucocorticoid-stimulated expression (S7B Fig and S3 and S8 Tables). At 71 T2D- or glucose-associated variants we further observed evidence for association with target gene expression (eQTL) in islets (S8 Table); for example, rs1107376 was an islet eQTL for *NPY* (P = 2.2x10⁻²¹).

At the 2p21 locus associated with glucose level, lead variant rs12712928 (BBJ beta = .068, P = 7.4x10⁻⁴⁶) mapped in a chromatin site with increased activity in glucocorticoid signaling and was proximal to *SIX2* and *SIX3* which both had glucocorticoid-induced expression (Fig 4C and 4D and S8 Table). This variant had the highest posterior probability in fine-mapping data (PPA = .89), suggesting it is likely causal for glucose association at this locus. This variant also had evidence for T2D association in BBJ (beta = .048, P = 2.1x10⁻⁶) and DIAMANTE (beta = .022, P = .012), and was the lead variant at a T2D signal recently reported in East Asians (P = 1.8x10⁻¹⁴) [42]. We therefore tested whether rs12712928 affected enhancer activity using sequence around variant alleles in untreated and dexamethasone treated MIN6 cells (see Methods). The glucose increasing and T2D risk allele C had significantly reduced enhancer activity in both glucocorticoid-treated (T-test P = 2.5x10⁻⁶) and untreated cells (T-test P = 3.2x10⁻⁴) (Fig 4E). However, the allelic differences at this variant were more pronounced in glucocorticoid-treated cells (ref/alt ratio GC = 6.85, 95% CI = 3.4,10.2; untreated = 1.78, 95% CI = 1.23,2.32, permutation test P = 5.1x10⁻³) (Fig 4F). We also observed evidence that rs10168523 was an islet eQTL for *SIX3* and *SIX2* (*SIX3* P = 5.1x10⁻²³, *SIX2* P = 8.2x10⁻¹⁰; Fig 4G), where

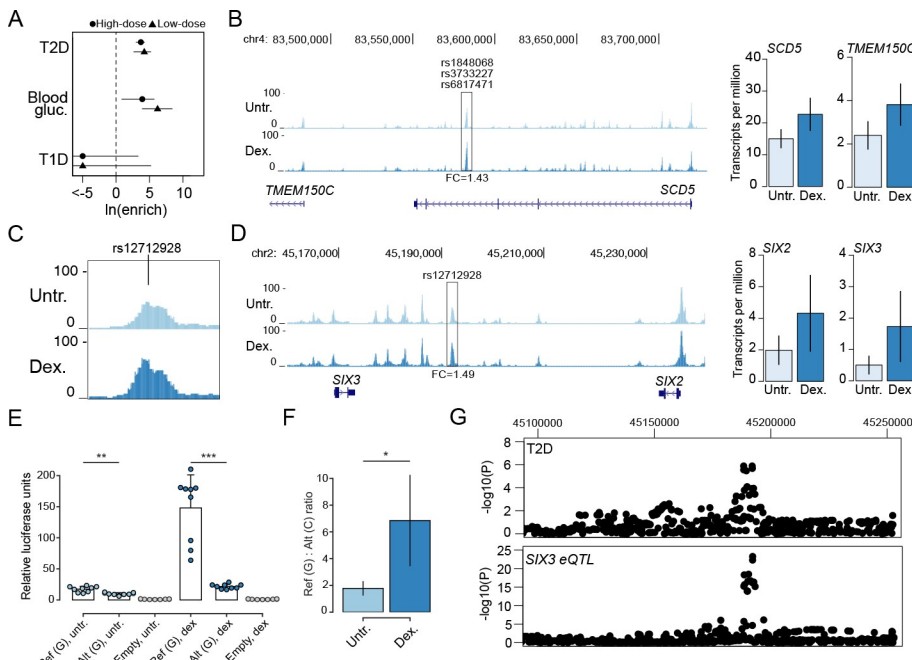

**Fig 4. Type 2 diabetes and glucose associated variants affect glucocorticoid-responsive islet regulatory programs.**
(A) Enrichment of variants associated with type 1 diabetes (T1D), type 2 diabetes (T2D) and blood sugar (glucose)
levels for differential chromatin sites in high-dose and low-dose glucocorticoid-treated islets. Values represent log
enrichment estimates and 95% confidence intervals. (B) Multiple fine-mapped T2D variants at the *SCD5/TMEM150C*
locus mapped in a glucocorticoid-responsive islet accessible chromatin site. Both the *SCD5* and *TMEM150C* genes had
increased expression in glucocorticoid-treated islets. Genome browser tracks represent RPKM normalized ATAC-seq
signal, and bar plots represent mean expression and standard error. (C, D) Variant rs12712928 with evidence for blood
sugar and T2D association mapped in a glucocorticoid-responsive chromatin site at the *SIX2/3* locus. Both the *SIX2*
and *SIX3* genes had increased expression in glucocorticoid-treated islets. Genome browser tracks represent RPKM
normalized ATAC-seq signal, and bar plots represent mean expression and standard error. (E) Variant rs12712928 had
significant allelic effects on enhancer activity in gene reporter assays in MIN6 cells. Values represent mean and
standard deviation. (F) The allelic effects of rs12712928 were more pronounced in glucocorticoid-treated relative to
untreated islets. Values represent fold-change and 95% CI. (G) The T2D association signal at *SIX2/3* was colocalized
with an eQTL for *SIX3* expression in islets. For panels B, C and D the values shown are from results using high-dose
treatment. For panels B and D, the fold-change (FC) in accessible chromatin signal in glucocorticoid treatment
compared to untreated is indicated at highlighted sites. ***$P < 1 \times 10^{-4}$, **$P < 1 \times 10^{-3}$, *$P < 1 \times 10^{-2}$.

the T2D risk allele was correlated with reduced expression of both genes. Glucose level and
T2D association at this locus was strongly co-localized with the *SIX3* and *SIX2* eQTLs (BBJ
T2D shared *SIX3* PP = 89%, *SIX2* PP = 98%; BBJ blood sugar shared *SIX3* PP = 98%, *SIX2*
PP = 99%) (Fig 4G).

These results reveal that variants associated with T2D and glucose level are enriched in glu-
cocorticoid-responsive chromatin sites in islets, including variants such as rs12712928 at the
*SIX2/3* locus which interact with glucocorticoid signaling directly to affect islet regulation.

## Discussion

Our study demonstrates the relevance of islet chromatin dynamics in response to corticoste-
roid signaling to T2D pathogenesis, including T2D risk variants that interact with corticoste-
roid activity directly to affect islet chromatin. In a similar manner, variants mediating
epigenomic responses of pancreatic islets to proinflammatory cytokines were recently shown
to contribute to genetic risk of T1D [34]. Numerous environmental signals and external condi-
tions modulate pancreatic islet function and contribute to the pathophysiology and genetic

basis of diabetes, yet the epigenomic and transcriptional responses of islets to disease-relevant stimuli have not been extensively measured. Future studies of islet chromatin and gene regulation exposed to additional stimuli will therefore likely continue providing additional insight into diabetes risk.

Glucocorticoid signaling led to broad changes in accessible chromatin, which up-regulated the expression of proximal genes enriched for processes related to ion channels and transport, in particular potassium channels. Potassium ion concentrations modulate calcium influx and insulin secretion in beta cells [43], and in disruption of ion channel function leads to impaired glucose-induced insulin secretion and diabetes [44]. Glucocorticoids have been shown to suppress calcium influx while preserving insulin secretion via cAMP [7], and in line with this finding we observed evidence for increased activity of potassium channel and cAMP signaling genes and decreased activity of phosphodiesterase genes. Up-regulated genes were also strong enriched in lipid metabolism pathways, which has been shown to regulate insulin secretion and contribute to diabetes [45,46]. Several up-regulated genes *PER1* and *CRY2* are also components of the circadian clock, and previous studies have shown that endogenous glucocorticoid release is under control of circadian rhythms and therefore may contribute to downstream regulation of the clock [47]. Conversely, glucocorticoid signaling down-regulated inflammatory programs, in line with previous reports and the known function of glucocorticoids [2,17,48], as well as key genes involved in islet function such as *NKX6-1*, *PAX6*, *RFX6*, and *ISL1*. Our findings further suggest that down-regulation of gene activity in glucocorticoid signaling is mediated through the activity of STAT and other TFs at proximal accessible chromatin sites, either through reduced TF expression or inhibition by GR. We also observed enrichment of FOXA binding in sites both up- and down-regulated in glucocorticoid signaling, suggesting these TFs mark sites that are broadly responsive to signal-dependent TF activity in islets in line with their known function as pioneer factors.

Genetic variants near the homeobox TFs *SIX2* and *SIX3* influence glucose levels [49,50], and our results provide evidence that both of these TFs operate downstream of glucocorticoid signaling and that the variants interact with this signaling program directly to influence glucose levels and risk of T2D. A previous study identified association between this locus and glucose levels in Chinese samples and demonstrated allelic effects of the same variant on islet enhancer activity and binding of the TF GABP [50], further supporting the likely causality of this variant. *SIX2* and *SIX3* have been widely studied for their role in forebrain, kidney and other tissue development [51–56]. In islets, both *SIX2* and *SIX3* have been shown to increase expression in adult compared to juvenile islets, and induction of *SIX3* expression in EndoC-βH1 cells and juvenile islets enhanced islet function, insulin content and secretion and may contribute to the suppression of proliferative programs [57]. In line with this finding, the glucose-lowering and T2D protective allele of the likely causal variant increased islet enhancer activity and *SIX2/3* expression.

Our *in vitro* experimental model mimics the environment of pancreatic islets under hormone signaling, albeit for a small number of treatments and conditions. Given the similarity in binding motifs of many nuclear hormone receptors and the enrichment of glucocorticoid responsive sites for androgen and progesterone receptor motifs, the effects of GR on islet gene regulation may overlap with other nuclear receptors by acting on shared chromatin sites [58]. Studies of other tissues have profiled glucocorticoid signaling across a broader range of experimental conditions and identified dose- and temporally-dependent effects on gene regulatory programs [14,15], and in islets dose- and temporally-dependent effects of glucocorticoids may impact insulin secretion and other islet functions. Future studies profiling the genomic activity of nuclear receptors in islets across a greater breadth of experimental conditions will therefore help further shed light into the role of hormone signaling dynamics in islet gene regulation and diabetes pathogenesis.

## Methods

### Ethics statement

All studies were approved by the Institutional Review Board of the University of California San Diego.

### Human islet samples

Human islet samples were obtained through the Integrated Islet Distribution Program (IIDP), University of Alberta and Prodo labs. Islet samples were further enriched using a dithizone stain. Islets were cultured for 24hr at approximately 10mL media/1k islets in 10cm dishes at 37C, 5% CO2 in CMRL 1066 media supplemented with 10% FBS, 1X pen-strep, 8mM glucose, 2mM L-glutamine, 1mM sodium pyruvate, 10mM HEPES, and 250ng/mL Amphotericin B. Treated islets had dexamethasone (Sigma) added in the culture media at either 100 ng/mL for 24hr, 4ng/mL for 24hr or 4 ng/mL for 6hr.

### ATAC-seq assays

Islet samples were collected and centrifuged at 500xg for 3 minutes, then washed twice in HBSS, and resuspended in nuclei permeabilization buffer consisting of 5% BSA, 0.2% IGE-PAL-CA630, 1mM DTT, and 1X complete EDTA-free protease inhibitor (Sigma) in 1X PBS. Islets were homogenized using a chilled glass dounce homogenizer and incubated on a tube rotator for 10 mins before being filtered through a 30uM filter (sysmex) and centrifuged at 500xg in a 4C microcentrifuge to pellet nuclei. Nuclei were resuspended in Tagmentation Buffer (Illumina) and counted using a Countess II Automated Cell Counter (Thermo). Approximately 50,000 nuclei were transferred to a 0.2mL PCR tube and volume was adjusted to 22.5uL with Tagmentation Buffer. 2.5uL TDE1 (Illumina) was added to each tagmentation reaction and mixed with gentle pipetting. Transposition reactions were incubated at 37C for 30 minutes. Tagmentation reactions were cleaned up using 2X reaction volume of Ampure XP beads (Beckman Coulter) and eluted in 20uL Buffer EB (Qiagen). 10uL tagmented DNA prepared as described above was used in a 25uL PCR reaction using NEBNext High-Fidelity Master Mix (New England Biolabs) and Nextera XT Dual-Indexed primers (Nextera). Final libraries were double size selected using Ampure XP beads and eluted in a final volume of 20uL Buffer EB. Libraries were analyzed using the Qubit HS DNA assay (Thermo) and Agilent 2200 Bioanalyzer (Agilent Biotechnologies). Sample libraries were sequenced on Illumina HiSeq 4000 using 100bp paired-end reads except for samples Isl10, Isl11 and Isl12 which were sequenced on Illumina NovaSeq 6000 using 100bp paired-end reads.

### RNA-seq assays

RNA was isolated from treated and untreated islets using RNeasy Mini kit (Qiagen) and submitted to the UCSD Institute for Genomic Medicine to prepare and sequence ribodepleted RNA libraries. Sample libraries were sequenced on Illumina HiSeq4000 using 100bp paired-end reads except for samples Isl10, Isl11 and Isl12 which were sequenced on Illumina NovaSeq 6000 using 100bp paired-end reads.

### ATAC-seq data processing

We trimmed reads using Trim Galore with options '–paired' and '–quality 10', then aligned them to the hg19 reference genome using BWA [59] mem with the '-M' flag. We then used samtools [60] to fix mate pairs, sort and index read alignments, used Picard (http://broadinstitute.github.io/picard/) to mark duplicate reads, and used samtools [60] to filer reads

with flags '-q 30', '-f 3', '-F 3332'. We then calculated the percentage of mitochondrial reads and percentage of reads mapping to blacklisted regions and removed all mitochondrial reads. We calculated a TSS enrichment score for each ATAC-seq experiment using the Python package 'tssenrich'. To obtain read depth signal tracks, we used bamCoverage [61] to obtain bigWig files for each alignment with signal normalization using RPKM.

### Identifying differential chromatin sites

We first used Irreproducible Discovery Rate (IDR) to define a set of canonical ATAC-seq sites for differential analysis. In brief, for each condition separately, we pooled reads across all assays and randomly split the pooled reads into two 'pseudo-replicates'. For the pooled and 'pseudo-replicate' data we called candidate peaks using MACS2 [62] with the parameters '—extsize 150 –keep-dup all–shift -75 –nomodel -p 0.01'. We applied IDR to the 'pseudo-replicate' candidate peak calls and obtained the number of peaks at an IDR threshold of .01. We then sorted and filtered the pooled candidate peak calls based on this number. Finally, we merged the resulting peaks across conditions, where if two peaks overlapped, we retained the more significant peak, and considered these canonical sites for downstream analyses.

The set of alignments for each assay were then supplied as inputs to the R function feature-Counts from the Rsubread [63] package to generate a matrix of read counts within each canonical site. We applied the R function DESeqDataSetFromMatrix from the DESeq2 [64] package to the read count matrix with default parameters then applied the DESeq function including donor as a variable to model paired samples. We considered sites differentially accessible with FDR<0.1, as computed by the Benjamini-Hochberg method.

We determined the percentage of differential sites with increased activity in glucocorticoids that overlapped a site active in untreated samples, as well as the percentage of differential sites proximal to a gene promoter defined as 5kb upstream of the transcription start site.

### Principal components analysis

We first defined input sites by merging overlapping (1bp or more) peaks identified in at least two experiments across all ATAC-seq experiments. We then constructed a read count matrix using edgeR [65] and calculated normalization factors using the 'calcNormFactors' function. We applied the voom transformation [66] and used the 'removeBatchEffect' function from limma [67] to regress out batch effects and sample quality effects (using TSS enrichment as a proxy for sample quality). We then restricted the read count matrix to the 100,000 most variable peaks and performed PCA analysis using the core R function 'prcomp' with rank 2.

### TF enrichment analysis

Differentially accessible chromatin sites were analyzed for sequence motif enrichment compared to a background of all chromatin sites tested for differential activity using HOMER [68] and a masked hg19 reference genome with the command 'findMotifsGenome.pl <bed file> <masked hg19> <output dir> -bg <background bed file> -size 200 -p 8 -bits -preparse -pre-parsedDir tmp'. We used the TF sequence motif database provided with the HOMER software. For TF ChIP-seq enrichment, we obtained ChIP-seq binding sites for 160 TFs generated by the ENCODE project [69] and tested for enrichment of binding in differential accessible chromatin sites compared to a background of all remaining chromatin sites genome-wide without differential activity. For each TF we calculated a 2x2 contingency table of overlap with differential sites and non-differential sites, determined significance using a Fisher test and calculated a fold-enrichment of overlap in differential compared to non-differential sites.

## RNA-seq data processing and analysis

Paired-end RNA-Seq reads were aligned to the genome using STAR [70] (2.5.3a) with a splice junction database built from the Gencode v19 gene annotation [71]. Gene expression values were quantified using the RSEM package (1.3.1) and filtered for >0.1 TPM on average per sample. Raw expression counts from the remaining 20,480 genes were normalized using variance stabilizing transformation (vst) from DESeq2 [64] and corrected for sample batch effects using limma removeBatchEffect. Principal component analysis was performed in R using the prcomp function. To identify differentially expressed genes between treated and untreated samples we obtained raw expression counts from RSEM [72] for the 20,480 genes and applied DESeq2 [64] with default settings including donor as a cofactor to model paired samples. To identify enriched GO terms in up and down-regulated genes, we applied GSEA [73] using Gene Ontology terms and KEGG/REACTOME pathway terms. We excluded gene sets with large numbers of genes in enrichment tests.

## Proximity of differential chromatin sites to differentially expressed genes

We calculated the percentage of differential accessible chromatin sites mapping within 100kb of (i) all differentially expressed genes, (ii) up-regulated genes and (iii) down-regulated genes compared to non-differentially accessible sites, and determined the significance and odds ratio using a Fisher exact test. We calculated a relative distance metric with bedtools [74] (reldist function) using either differential chromatin sites or a background of all islet accessible chromatin sites as the "a" argument and differentially expressed genes as the "b" argument. We compared the distribution of relative distances from differential sites to the distribution from background sites using a Kolmogorov-Smirnov test.

## Sample genotyping and imputation

Non-islet tissue was collected for seven samples during islet picking and used for genomic DNA extraction using the PureLink genomic DNA kit (Invitrogen). Genotyping was performed using Infinium Omni2.5–8 arrays (Illumina) at the UCSD Institute for Genomic Medicine. We called genotypes using GenomeStudio (v.2.0.4) with default settings. We then used PLINK [75] to filter out variants with 1) minor allele frequency (MAF) less than 0.01 in the Haplotype Reference Consortium (HRC) [76] panel r1.1 and 2) ambiguous A/T or G/C alleles with MAF greater than 0.4. For variants that passed these filters, we imputed genotypes into the HRC reference panel r1.1 using the Michigan Imputation Server with minimac4. Post imputation, we removed imputed genotypes with low imputation quality ($R2 < .3$).

## Allelic imbalance mapping

We identified heterozygous variant calls in each sample with read depth of at least 10 in both untreated and treated cells, and then used WASP [77] to correct for reference mapping bias. We retained variants in each sample where both alleles were identified at least 3 times across untreated and treated cells. We then merged read counts at heterozygous SNPs from all samples in untreated and treated cells separately. We fit a beta-binomial model to the observed allele counts using the method of NPBin [78]. The parameters of the beta-binomial model were $\alpha = 40.78$ and $\beta = 39.26$ with over-dispersion of .012 for untreated samples and $\alpha = 41.76$ and $\beta = 40.10$ with over-dispersion of .012 for glucocorticoid-treated samples. We called imbalanced variants from the merged counts using a beta-binomial test, and then calculated q-values from the resulting beta-binomial p-values. We considered variants significant at FDR < .10.

## Heterogeneous allelic imbalance

For all variants with significant allelic imbalance in either glucocorticoid-treated or untreated conditions, we tested for heterogeneity in imbalance between conditions. We used Pearson's chi-squared test as implemented in the "prop.test" function of R. We calculated q-values from the resulting p-values and considered variants significant at FDR < .10.

## Genetic association analysis

We tested glucocorticoid-responsive chromatin sites for enrichment of diabetes association using genome-wide association data for T1D [79], T2D from the DIAMANTE consortium [22], and blood sugar (glucose) from the Japan Biobank study [49]. For each study we retained variants with minor allele frequency (MAF)>.05 and tested for enrichment of high-dose and low-dose differential sites using fgwas [40] with a window size of 1Mb.

We then cataloged all variants in glucocorticoid-responsive chromatin sites in T2D and glucose fine-mapping data and with nominal association (P < .005) genome-wide. For DIAMANTE, we used fine-mapping results provided with the study. For the Japan Biobank, we fine-mapped signals ourselves using summary statistics. We calculated approximate Bayes factors (ABF) for each variant as described previously [80]. We compiled index variants for each locus and defined variants within a 5 Mb window and at least low linkage ($r^2$>0.1) in the East Asian subset of 1000 Genomes [81] with each index. For each variant, we calculated posterior probabilities of associations (PPA) by dividing the variant ABF by the sum of ABF for the locus. We defined 99% credible sets by sorting variants by descending PPA and retaining variants up to a cumulative probability of 99%. For each variant in glucocorticoid-responsive chromatin, we identified protein-coding genes in GENCODE v33 with differential expression and where the gene body mapped within 100kb of the variant.

## Expression QTL analyses

We obtained islet expression QTL data from a published study [82]. We extracted variant associations at the *SIX2/SIX3* locus and tested for colocalization between T2D and blood sugar association in the Biobank Japan study and *SIX2* and *SIX3* eQTLs using a Bayesian approach [83]. We considered signals colocalized with shared PP greater than 80%.

## Gene reporter assays

To test for allelic differences in enhancer activity at the *SIX2/3* locus, we cloned human DNA sequences (Coriell) containing the reference allele upstream of the minimal promoter in the luciferase reporter vector pGL4.23 (Promega) using the enzymes Sac I and Kpn I. A construct containing the alternate allele was then created using the NEB Q5 SDM kit (New England Biolabs). The primer sequences used were as follows:

- Cloning FWD AGCTAGGTACCCCTCATCTGCCTTTCTGGAC

- Cloning REV TAACTGAGCTCCAGTGGGTATTGCTGCTTCC

- SDM FWD TGCATTGTTTcCTGTCCTGAAGACGAGC

- SDM REV GGGGGTGCCTGCATCTGC

MIN6 cells were seeded at approximately 2.5E05 cells/cm^2 into a 48-well plate. The day after passaging into the 48-well plate, cells were co-transfected with 250ng of experimental firefly luciferase vector pGL4.23 containing the alt or ref allele in the forward direction or an empty pGL4.23 vector, and 15ng pRL-SV40 Renilla luciferase vector (Promega) using the

Lipofectamine 3000 reagent. Cells were fed culture media and stimulated where applicable 24 hours post-transfection. For stimulation 100 ng/mL dexamethasone (Sigma) was added to the culture media. Cells were lysed 48 hours post transfection and assayed using the Dual-Luciferase Reporter system (Promega). Firefly activity was normalized to Renilla activity and normalized results were expressed as fold change compared to the luciferase activity of the empty vector. The python package 'luciferase' was then used to remove batch effects. A two-sided t-test was used to compare the luciferase activity between the two alleles or between treatments. A permutation test was used to compare the allelic ratio of luciferase activity between the two treatments, based on 100,000 permutations of the allele labels.

## Supporting information

**S1 Fig. Gene expression in islets in response to different doses and durations of glucocorticoid treatment.** Expression level of (A) *ZBTB16*, (B) *VIPR1* and (C) *IL11* in high-dose (100ng/mL for 24hr), low-dose (4ng/mL for 24hr or 6hr) glucocorticoid-treated or untreated islets. Values represent mean expression and standard error.
(TIF)

**S2 Fig. Islet accessible chromatin signal across replicate samples at *ZBTB16*.** RPKM normalized ATAC-seq signal for individual islet sample in high-dose glucocorticoid treated and untreated islets. Sites with differences in chromatin accessibility across conditions are highlighted.
(TIF)

**S3 Fig. Islet accessible chromatin signal across replicate samples at *VIPR1*.** RPKM normalized ATAC-seq signal for individual islet sample in high-dose glucocorticoid treated and untreated islets. Sites with differences in chromatin accessibility across conditions are highlighted.
(TIF)

**S4 Fig. Accessible chromatin signal in islets in response to low dose glucocorticoid treatment.** RPKM normalized ATAC-seq signal in low-dose (4ng/mL for 6hr) glucocorticoid treated and untreated islets at the (A) *ZBTB16* and (B) *VIPR1* loci. Sites induced by glucocorticoid treatment are highlighted.
(TIF)

**S5 Fig. Islet accessible chromatin signal at *IL11*.** RPKM normalized ATAC-seq signal in high-dose glucocorticoid treated and untreated islets at the *IL11* locus. The *IL11* promoter which has reduced accessibility in glucocorticoid treated islets at high dose is highlighted.
(TIF)

**S6 Fig. Differential chromatin accessibility in high- and low-dose glucocorticoid treatment.** (A) Venn diagram of overlap in sites with differential activity in high-dose (100ng/mL for 24hr, n = 6) and low-dose (4ng/mL for 6hr, n = 3) glucocorticoid treatment. (B) Effects of high-dose and low-dose glucocorticoid treatment on sites with significant differential activity in either treatment.
(TIF)

**S7 Fig. Differential gene expression in high- and low-dose glucocorticoid treatment.** (A,B) Volcano plot of differential gene expression in glucocorticoid-treated islets at low dose for 24hr or 6hr compared to untreated islets. Genes with significant differential expression (FDR < .10) are highlighted in red, and genes with most pronounced changes in expression are

listed. (C) Venn diagram of overlap between genes differentially expressed in 24hr high
(n = 6), 24hr low (n = 3), 6hr low (n = 3) glucocorticoid treatment. (D) Effects of 24hr high-
and low-dose treatment on genes with significant differential expression in either treatment.
(E) Effects of 24hr high- and 6hr low-dose treatment on genes with significant differential
expression in either treatment.
(TIF)

**S8 Fig. T2D-associated variants in differential chromatin sites.** (A) Multiple variants at the
*FBXW7/TMEM154* locus mapped in a site with decreased activity and *FBXW7* had decreased
expression in glucocorticoid stimulation. (B) A variant at the *NPY* locus mapped in a site with
increased activity and NPY had increased expression in glucocorticoid stimulation. Genome
browser tracks represent RPKM normalized ATAC-seq signal, and expression bar plots repre-
sent mean expression and standard error. Values shown are from high-dose treatment. The
fold-change (FC) in accessible chromatin signal in glucocorticoid treatment compared to
untreated is indicated at highlighted sites.
(TIF)

**S1 Table. Human islet donor samples.** Islet samples used for genomic assays in this study and
donor characteristics.
(XLSX)

**S2 Table. Islet accessible chromatin sites.** Complete list of 127,228 reproducible islet accessi-
ble chromatin sites identified by IDR.
(XLSX)

**S3 Table. Islet chromatin sites with differential activity in glucocorticoid treatment.** List of
islet accessible chromatin sites with differential activity in each treatment dose and duration
using DESeq2.
(XLSX)

**S4 Table. TFs enriched in differential chromatin sites.** Sequence motifs and TF binding sites
enriched in differential islet accessible chromatin sites in each treatment dose and duration.
(XLSX)

**S5 Table. Genetic variants with allelic imbalance in islet chromatin.** List of variants with sig-
nificant effects on accessible chromatin in untreated or glucocorticoid treated islets.
(XLSX)

**S6 Table. Genes with differential expression in glucocorticoid-treated islets.** Genes with dif-
ferential expression in each treatment dose and duration using DESeq2.
(XLSX)

**S7 Table. Gene sets enriched in glucocorticoid-treated islets.** Gene ontology and pathway
terms enriched among genes with differential expression in each treatment dose and duration
using GSEA.
(XLSX)

**S8 Table. Diabetes risk variants in islet glucocorticoid-responsive chromatin sites.** Genetic
variants in 99% credible sets from fine-mapping data or with nominal association in genome-
wide summary statistic data from the DIAMANTE and Japan Biobank studies that mapped in
islet accessible chromatin sites with differential activity.
(XLSX)

**S1 Data. Source data.** Data underlying Figs 2F and 4E in the manuscript.
(XLSX)

## Acknowledgments

We thank the UC San Diego IGM Genomics Center for sequencing and technical support.

## Author Contributions

**Conceptualization:** Kyle J. Gaulton.

**Data curation:** Anthony Aylward, Mei-Lin Okino, Paola Benaglio, Joshua Chiou, Elisha Beebe, Jose Andres Padilla, Sharlene Diep.

**Formal analysis:** Anthony Aylward, Mei-Lin Okino, Paola Benaglio, Joshua Chiou, Elisha Beebe, Kyle J. Gaulton.

**Funding acquisition:** Kyle J. Gaulton.

**Investigation:** Mei-Lin Okino, Jose Andres Padilla, Sharlene Diep.

**Methodology:** Anthony Aylward, Paola Benaglio, Joshua Chiou, Elisha Beebe.

**Resources:** Mei-Lin Okino.

**Software:** Anthony Aylward.

**Supervision:** Kyle J. Gaulton.

**Writing – original draft:** Anthony Aylward, Mei-Lin Okino, Paola Benaglio, Kyle J. Gaulton.

**Writing – review & editing:** Anthony Aylward, Mei-Lin Okino, Paola Benaglio, Joshua Chiou, Kyle J. Gaulton.

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
