## [Decision Letter · Decision Letter 0]

7 Aug 2020

Dear Dr Gaulton,

Thank you very much for submitting your Research Article entitled 'Glucocorticoid signaling in pancreatic islets modulates gene regulatory programs and genetic risk of type 2 diabetes' to PLOS Genetics. Your manuscript was fully evaluated at the editorial level and by independent peer reviewers. The reviewers appreciated the attention to an important problem, but raised some substantial concerns about the current manuscript. Based on the reviews, we will not be able to accept this version of the manuscript, but we would be willing to review again a much-revised version. We cannot, of course, promise publication at that time.

If you decide to revise the manuscript for further consideration at PLOS Genetics, please aim to resubmit within the next 60 days, unless it will take extra time to address the concerns of the reviewers, in which case we would appreciate an expected resubmission date by email to plosgenetics@plos.org.

[LINK]

We are sorry that we cannot be more positive about your manuscript at this stage. Please do not hesitate to contact us if you have any concerns or questions.

Yours sincerely,

Michael L Stitzel, PhD

Guest Editor

PLOS Genetics

Wendy Bickmore

Section Editor: Epigenetics

PLOS Genetics

Overall, this is an interesting study that could provide potential new insights into transcriptional regulation of steroid responses in islets and nominate potential molecular effects of type 2 diabetes-associated genetic variants, which makes it of potential interest to the PLOS Genetics readership and, in principle, appropriate and suitable for publication in PLOS Genetics pending revisions addressing reviewer concerns.

Technical concerns raised by two reviewers related to analyses are valid, and should be straightforward for the authors to address and rectify. The broader and more major concern relates to the dose and duration of the dexamethasone treatment, which seems higher than that to which the islets would be exposed in vivo in physiologic or even pathophysiologic states. In particular, the authors need to consider the physiological relevance of the dose of dexamethasone they are using and the rather late time point (24 hours). Direct physiological targets of glucocorticoids respond very rapidly (within 10 mins) and to low concentrations. With high doses for long periods there will be a lot of noise from indirect targets, and some genes (or regulatory elements) induced acutely early in the response will have returned toward baseline by 24 hrs. Data indicating that the epigenetic and gene expression changes described in this study hold true in response to more physiologically or medically relevant steroid levels are necessary to solidify the specificity and validity of the observations currently described and to make the study suitable for publication in PLOS Genetics.

In addition to the reviewer’s comments, we request the authors to deal with ATAC-seq peaks between replicates using IDR (this is the gold standard), not by simply merging peaks.

" ext-link-type="uri" xlink:type="simple">https://www.encodeproject.org/atac-seq/"

Reviewer's Responses to Questions

**Comments to the Authors:**

Reviewer #1: In this manuscript, Aylward et al identified how dexamethasone-induced glucocorticoid signaling in primary islet cells affect transcription by studying the changes in accessible chromatin and gene expression with ATACseq and RNAseq. First they discovered that ~3,000 genomic regions and ~1,000 genes responded to Dex treatment. The genomic regions were enriched for glucorticoid response elements and genes were enriched for steroid and lipid metabolism as well as ion transport. They then identified type 2 diabetes and glucose-associated genetic variants that overlap the Dex-responsive genomic regions and identified variants that had allelic imbalance in chromatin accessibility and had an effect on gene expression. Overall, this ia very nice study with interesting and well-presented findings. The conclusions are supported by the data and the results are nice interpreted. I have only a few minor suggestions that are mostly grammar errors, etc.

1. Line 181: remove "both"

2. Add H to indicate where the panel H is in Figure 2.

3. Line 308: "These results demonstrate that T2D and glucose level variants are enriched..." I do not know if the results show enrichment. It certainly shows presence but of all the T2D-associated SNPs, is there enrichment in the GR-responsive sites? You can either do an enrichment analysis or rephrase this claim.

4. Fig 2 caption - Line 569: remove x, i think you mean 6 donors.

5. Fig 2 caption - Line 571: It is not clear what you mean by average values? Is is the average value of the loadings of the principal components?

6. In Fig 2G/H, it seems that rs684374 is a quadallelic SNP. Therefore, please identify which allele you consider to be the ref allele (G?) and which one you consider to be the alt allele (C?) on the figure.

7. Fig 4e/f, please identify which allele is ref and which is alt on the figure.

Reviewer #2: In this work, Aylward et al. investigate how glucocorticoid dexamethasone treatment affects chromatin accessibility and gene expression in cultured islet cells by performing RNA-seq and ATAC-seq. The authors identify differentially accessible sites and differentially expressed genes, enriched for pathways relevant to glucocorticoid signaling and inflammatory/stress response. The authors then use these profiles to fine map T2D GWAS loci. The study design is reasonable and the work could be quite useful to the community. I have the following comments:

1. To obtain the consensus set of ATAC-seq peaks across conditions and replicates, the authors merged peaks called in individual samples. Were sequencing depth differences in each replicate accounted for in this process? Eg. peaks could be identified in a particular condition owing to just higher depth, these would go into the final peak set and could be identified as differentially expressed?

2. Fig 2B shows the number of upregulated or downregulated accessible chromatin sites under dex vs untreated condition. A more informative figure could show for each peak tested, effect sizes/fold changes and p values to better appreciate the spread of the data. On similar lines of Fig 2C etc.

3. It is confusing why examples for motifs enriched up or downregulated noted in the main text (lines 151-156) are not all labeled in Fig 2C. I then realized that the panels 2C and 2D are flipped in the main text vs figure.

4. In the examples mentioned in lines 151-156, FOXA1 is noted to be enriched in upregulated peaks whereas FOXA2 is noted to be enriched in downregulated peaks. Checking the JASPAR motif database, the motifs for FOXA1 and 2 are largely similar, so it is unclear what to interpret from these results. What motif database was used for this? Maybe there should be a fold change threshold applied, since enrichment p values could just be driven by the motif PWM information content.

5. Differing from other places in the text, line 268 mentions P values in the -log10 scale. These values are barely significant and it’s hard to interpret/appreciate the enrichment. Looks like the background set of ATAC peaks were sampled from ATAC peaks across all experiments. A comparison between differentially accessible sites and unaltered sites could be more appropriate.

6. For fine-mapped variants in differentially accessible sites, the authors nominated target genes using proximity and differential gene expression. As later shown in the manuscript for one locus, how many of the 412 variants have evidence of association with gene expression from islet eQTL data?

Minor comments:

1. Line 123 is referring to chromatin accessibility so the intended figure reference was liekly 2D,E instead of “2B,C”. Also, panels are mislabeled in in the Fig 1 legend (lines 560-566).

2. Fig 1B and C show ‘untr’ first and then ‘dex’. For consistency, in Figs D and E could also show ‘untr’ first and then ‘dex’. Also, track colors in Figs 1D and E could be made consistent with colors in Fig 1 B and C.

3. Fig 1E, seems that the highlighted peak in ‘dex’ goes higher than the range 100. The differences could be clearer if the y ranges are extended in these tracks.

4. Line 133 mentions that the PCA was performed on “read counts” which sounds like no normalization was performed. The methods section then notes that normalization was indeed performed. The main text should be clarified to avoid confusion.

5. In figures 1D, 1E and 2E, labeling a quantitative measure of the differences in the peak signals, i.e. fold changes from the differential analyses would be informative, on top of appreciating the differences qualitatively.

6. Figure 3D: what is the unit for distance on the x axis?

7. Line 289 - it would be informative to mention the blood glucose GWAS SNP OR and P value. Presumably this is the lead variant as it gets the highest PPA? Line 294 mentions the Japan Biobank T2D P values, it would be useful to mention P values, OR and PPA from Diamante T2D data as well.

Reviewer #3: The manuscript by Aylward and colleagues investigates gene expression and chromatin accessibility response to glucocorticoids in pancreatic islets. The manuscript is overall clear and logically builds through the different steps of analysis to demonstrate that 1. GC induce changes in gene expression and chromatin accessibility, 2. Which TFs are involved in this response, 3. That T2D risk variants are enriched in GC response chromatin accessibility regions and finally 4. Aims to provide experimental validation of GC-dependent allele-specific enhancer activity for a putative T2D causal variant.

Overall the paper is based on solid data, however a few points require further clarification.

Major points:

1. Allelic imbalance analysis. A binomial test is not appropriate for this analysis, as it does not account for over-dispersion. A beta-binomial distribution should be used.

2. Please report GSEA results after multiple-test correction.

3. Allele-specific enhancer activity at rs12712928. A formal test should be performed to compare the differential enhancer activity between the two alleles.

4. Differential chromatin accessibility analysis: How many peaks were tested for the differential chromatin accessibility analysis in DESEQ? Were the ATAC-seq data normalized prior to differential chromatin accessibility analysis? If they were not normalized, please repeat the analysis to reflect the same workflow used for differential gene expression.

Minor points:

1. Line 171, please specify what type of cells are MIN6.

2. Figure 3E. Please include a legend to interpret the dot size. What is the scale of the X axis and why all dots align in the same position?

3. Please report in the text the enrichment for T2D and glucose variants in GC-responsive chromatin.

4. What concentration of dexamethasone was used in the reporter gene assays? Please add this information in the methods.

Reviewer #4: Aylward and colleagues investigate the role of glucocorticoid signaling in pancreatic islets, using several high throughput sequencing analyses in human islet specimens, combined with analyses of publicly available datasets in order to determine causal regulatory profiles between GR signaling and T2D. Numerous steroid responsive genes are identified and confirmed after in vitro culture with dexamethasone. The authors identify genetic variants for glucose control and T2D risk overlaps with numerous corticosteroid responsive chromatin sites and focused on a highly specific causal link between diabetes risk and glucocorticoid signaling, driven by the key human beta cell genes SIX2 and SIX3. The authors conclude a role for glucocorticoid signaling in the genetic risk for T2D.

General comments:

The authors utilize multiple sophisticated sequencing and bioinformatics approaches to develop a regulatory map of glucocorticoid-responsive genes/chromatin in human pancreatic islets. Given the connections between Cushing’s syndrome and diabetes in humans, these results provide new results not previously available in human islets that may have bearing on both diseases. The study is well written and bioinformatics studies are thorough. However, there are several significant concerns, detailed below, with the experimental design of the study on a relative small number of islet samples could lead to conclusions that may not truly approximate the role of glucocorticoid signaling in beta cells and may lead to overinterpretation of the role of GR in the development of T2D.

Specific Comments:

1. A major concern is the use of dexamethasone at an exceedingly high dose of 100 ng/mL for 24 hours to profile glucocorticoid signaling. This dose is 25-fold higher than physiologic glucocorticoid doses and also dramatically higher than what are even observed in patients with Cushing’s syndrome. At this dose, it is highly likely that the results will lead the authors to interpret roles for glucocorticoids that would not reflect physiologic changes related to T2D in vivo. It is crucial for the authors to perform their studies at (1) physiologic glucocorticoid doses, (2) a dose response curve favoring lower glucocorticoid doses, or (3) in the presence of pharmacologic inhibitors to ensure their results are specific and not due to off target effects.

2. While the authors did confirm the activation of GR responsive genes, they also note numerous additional nuclear hormone receptors activated by dexamethasone (given enrichment for ARE and PGR sites). These effects are again likely related to supraphysiologic glucocorticoid levels. It would be crucial to determine if these sites are enriched with more appropriate glucocorticoid doses. Alternatively, these sites may represent a nexus of overlapping binding sites shared for glucocorticoid and other steroid hormones. Thus, the authors should consider treating islets with ligands for these additional steroid responsive hormone receptors to see if GR elements are similarly induced.

3. Several of the results are confusing as it is unclear if the authors infer whether glucocorticoid activity is favorable as “signaling increases the activity of genes involved in islet function and insulin secretion while suppressing inflammatory and proliferative gene activity”, or whether it is detrimental due to impairment of key ion channels. At the doses applied, it would be expected that beta cell function would be impaired as key beta cells genes including NKX6.1 are downregulated.

Minor Comments:

1. The legend for Figure 1 is mislabeled as Figure 1E is missing but 1C is listed twice.

2. What do error bars represent in their figures? This is not clear. SD, SEM?

**Have all data underlying the figures and results presented in the manuscript been provided?**

Reviewer #1: Yes

Reviewer #2: None

Reviewer #3: **No: **Processed data and annotations will be made available in https://www.diabetesepigenome.org upon publication. Data should also be uploaded on SRA.

Reviewer #4: Yes

PLOS authors have the option to publish the peer review history of their article (what does this mean?). If published, this will include your full peer review and any attached files.

Reviewer #1: No

Reviewer #3: No

Reviewer #4: No

---

## [Decision Letter · Decision Letter 1]

8 Feb 2021

Dear Dr Gaulton,

Thank you very much for submitting your Research Article entitled 'Glucocorticoid signaling in pancreatic islets modulates gene regulatory programs and genetic risk of type 2 diabetes' to PLOS Genetics.

The manuscript was fully evaluated at the editorial level and by independent peer reviewers. The reviewers appreciated the revisions that you have made but identified some remaining minor concerns that we ask you address in a revised manuscript

We therefore ask you to modify the manuscript according to the review recommendations. Your revisions should address the specific points made by each reviewer.

[LINK]

Yours sincerely,

Michael L Stitzel, PhD

Guest Editor

PLOS Genetics

Wendy Bickmore

Section Editor: Epigenetics

PLOS Genetics

Dear Dr. Gaulton and colleagues,

Congratulations! As you can see from the Reviewers' comments below, they found this revised submission much improved (and I agree). Based on Reviewer responses, we request the following minor revisions be made to facilitate publication of this study:

Reviewer 2 requested that the underlying profiling data be deposited on a public database such as NCBI Short Read Archive (SRA) in accordance with PLOS Genetics' policy for a few additional minor edits. Reviewer 3 requested two adjustments to the revised analyses and related text in the manuscript, namely that i) you report and describe only sites with evidence of significant imbalance after correcting for multiple testing, rather than those passing a nominal threshold of significance, and that formal testing should be applied to substantiate and support the identification and discussion of condition-specific allelic effects.

Pending satisfactory completion of these final minor revision requests, this manuscript will be suitable for publication in PLOS Genetics.

Warm Regards,

Michael

Reviewer's Responses to Questions

**Comments to the Authors:**

Reviewer #1: I had minor comments for the original submission. The authors satisfactorily addressed them.

Reviewer #2: The authors have made numerous edits and included additional experiments and analyses that all helped address my comments. I also find responses to other reviewer comments reasonable as well. These modifications have made the manuscript more robust. As promised, the authors should make sure to share all raw and processed data through appropriate channels.

Reviewer #3: The revised version of this manuscript addresses all my previous comments. The authors have done a great job in responding to the comments of all reviewers.

I only have two additional comments on the new and corrected analysis of allelic imbalance that is presented in this revised version. The results reported in the paragraph starting at line 205 should focus on sites with significant imbalance after multiple test correction, e.g. q value 0.1 (equivalent to 10% FDR). The manuscript is currently using a nominally significant p-value threshold of 0.05.

Additionally, if any inference is to be presented on condition-specific allelic imbalance, a formal test should be performed to contrast the allelic imbalance in the treatment and control conditions.

Reviewer #4: Authors have addressed my remaining concerns with the manuscript.

**Have all data underlying the figures and results presented in the manuscript been provided?**

Reviewer #1: Yes

Reviewer #2: None

Reviewer #3: Yes

Reviewer #4: None

PLOS authors have the option to publish the peer review history of their article (what does this mean?). If published, this will include your full peer review and any attached files.

Reviewer #1: **Yes: **Mete Civelek

Reviewer #2: No

Reviewer #3: No

Reviewer #4: No

---

## [Editor Report · Decision Letter 2]

6 Apr 2021

Dear Dr Gaulton,

We are pleased to inform you that your manuscript entitled "Glucocorticoid signaling in pancreatic islets modulates gene regulatory programs and genetic risk of type 2 diabetes" has been editorially accepted for publication in PLOS Genetics. Congratulations!

Yours sincerely,

Michael L Stitzel, PhD

Guest Editor

PLOS Genetics

Wendy Bickmore

Section Editor: Epigenetics

PLOS Genetics

Comments from the reviewers (if applicable):

**Data Deposition**

http://datadryad.org/submit?journalID=pgeneticsmanu=PGENETICS-D-20-00824R2

**Press Queries**

---

## [Editor Report · Acceptance letter]

10 May 2021

PGENETICS-D-20-00824R2 

Glucocorticoid signaling in pancreatic islets modulates gene regulatory programs and genetic risk of type 2 diabetes 

Dear Dr Gaulton, 

We are pleased to inform you that your manuscript entitled "Glucocorticoid signaling in pancreatic islets modulates gene regulatory programs and genetic risk of type 2 diabetes" has been formally accepted for publication in PLOS Genetics! Your manuscript is now with our production department and you will be notified of the publication date in due course.

With kind regards,

Katalin Szabo

PLOS Genetics

On behalf of:
